# Comparative Analysis of Meteorological versus In Situ Variables in Ship Thermal Simulations

**DOI:** 10.3390/s24082454

**Published:** 2024-04-11

**Authors:** Elena Arce, Andrés Suárez-García, José Antonio López-Vázquez, Rosa Devesa-Rey

**Affiliations:** 1Polytechnic School of Engineering of Ferrol, University of A Coruña, 15403 Ferrol, Spain; jose.lopez@udc.es; 2Defense University Center, Spanish Naval Academy, University of Vigo, 36920 Marín, Spain; andsuarez@uvigo.es; 3School of Industrial Engineering, University of Vigo, 36310 Vigo, Spain; rosa.devesa.rey@cud.uvigo.es

**Keywords:** low-cost weather station, meteorological variables, thermal simulation, ship energy efficiency

## Abstract

Thermal simulations have become increasingly popular in assessing energy efficiency and predicting thermal behaviors in various structures. Calibration of these simulations is essential for accurate predictions. A crucial aspect of this calibration involves investigating the influence of meteorological variables. This study aims to explore the impact of meteorological variables on thermal simulations, particularly focusing on ships. Using TRNSYS (TRaNsient System Simulation) software (v17), renowned for its capability to model complex energy systems within buildings, the significance of incorporating meteorological data into thermal simulations was analyzed. The investigation centered on a patrol vessel stationed in a port in Galicia, northwest Spain. To ensure accuracy, we not only utilized the vessel’s dimensions but also conducted in situ temperature measurements onboard. Furthermore, a dedicated weather station was installed to capture real-time meteorological data. Data from multiple sources, including Meteonorm and MeteoGalicia, were collected for comparative analysis. By juxtaposing simulations based on meteorological variables against those relying solely on in situ measurements, we sought to discern the relative merits of each approach in enhancing the fidelity of thermal simulations.

## 1. Introduction

Thermal modeling has become indispensable for predicting the energy requirements of buildings, yet the efficacy of these models under idealized conditions is often met with skepticism. Such simulations provide a digital lens through which the thermal dynamics of structures can be examined, facilitating the optimization of their design prior to and following construction. Ensuring the precision of these models is pivotal for achieving dependable outcomes [1,2,3,4].

The process of energy modeling considers the architecture and materials of a building, alongside external climatic conditions and the building’s usage patterns, including occupancy rates, operational timings, and internal loads stemming from its designated function [5]. The significance of energy calibration models has surged in recent times, heralding the potential for energy conservation through refined onsite measurements, thereby enabling more accurate energy assessments [6,7]. A growing body of research underscores the critical role of localized meteorological data in enhancing the fidelity of energy simulations, advocating for the integration of site-specific weather records to achieve accurately calibrated models.

The study of energy consumption in ships presents distinctive challenges necessitating the formulation of comprehensive mathematical or physical models to encapsulate the intricate interactions within and around the ship’s structure [8]. Internally, factors including construction attributes and HVAC systems are pivotal, while external elements such as weather conditions and infiltration patterns must also be considered. Furthermore, the integration of performance data, such as space temperature and humidity levels, within the vessel is imperative.

In contrast to land-based structures, ship simulations demand consideration of water temperature, adding complexity to their development. In the realm of thermal simulations for offshore applications, the availability and precision of meteorological data present notable hurdles. Ideally, historical meteorological datasets used for simulation periods should encompass all requisite variables and be tailored to the ship’s immediate environment. However, obtaining in situ weather data is often intricate, prompting researchers to explore alternative sources such as meteorological agencies. Consequently, the creation of in situ weather datasets, gathered hourly and encompassing all pertinent weather parameters, emerges as a fundamental prerequisite for accurately evaluating the energy demand of ships or constructions via thermal simulations.

While advanced thermal simulation techniques have been extensively studied and applied in the building sector, there remains a notable gap in research concerning their application in the maritime domain, particularly in ships. The existing literature predominantly focuses on energy conservation in ships for passenger or cargo transport [9,10,11], as well as in fishing vessels, driven by fluctuations in oil prices and the economic viability of maritime fleets. Additionally, there are studies within naval engineering that investigate thermal dynamics in propulsion systems [12].

This paper presents a study evaluating the performance of low-cost and ad-hoc sensors for the development of an energy calibration methodology within the TRNSYS simulation framework. The focus lies on assessing the relevance of model calibration for enhancing energy efficiency in ship design or retrofitting. TRNSYS, a modular simulation software tailored for transient system simulations, particularly in thermal zones, serves as the primary tool. A ship situated in Pontevedra, Galicia, a region characterized by complex orography and influenced by the Atlantic Ocean, serves as the study site. Through the comparison of data obtained from these sensors with observations from automatic weather stations operated by the Galician Meteorological Agency (MeteoGalicia), we explore the efficacy of meteorological variables against in situ measurements in thermal simulations [13]. This comparative analysis provides insights into the strengths and limitations of each approach, contributing to a better understanding of energy calibration methodologies in simulation environments.

The following sections present a general description of the developed station, followed by a description of the ship. The meteorological variables are presented, and the system used to evaluate and compare the performance of the two data sources is described. Finally, the results are presented and discussed, followed by a brief summary and conclusions.

## 2. Materials and Methods

### 2.1. Ship Description

The target ship, shown in Figure 1, is a military patrol. This patrol vessel, named Tabarca, belongs to the Ferrol Maritime Action Force Units Command. Propulsion is provided by a single “BAZÁN MTU” model engine (MTU, Friedrichshafen, Germany) generating 4500 horsepower, enabling a maximum speed of 15 knots. Additionally, it is equipped with two auxiliary diesel engines of CHRYSLER-BARREIROS type BS-36ME (Chrysler, Detroit, MI, USA) that operate at 1500 rpm with supercharging capabilities. Construction features include a hull crafted from 7 mm A-grade steel and internally insulated with a 5 cm layer of thermal insulation (MW40 mineral wool). The vessel is fully welded, measuring 44.5 m in length and 6.6 m in beam width. Given its construction date, enhancements could be necessary to improve its thermal efficiency.

The Tabarca patrol belongs to the Spanish Navy, and it is used for navigation and occasionally made available for research tasks. Its construction details are known, including its enclosures, which regulate the flow of external air, the incidence of solar radiation, the entry of natural light, and the transmission of heat (both inward and outward). The vessel’s materials and geometry are also documented, and it is constructed with fully welded A-grade naval steel measuring 44.5 m in length and 6.6 m in beam. The Tabarca’s hull consists of 7 mm thick A-grade steel coated internally with a 5 cm layer of thermal insulation known as rock wool. For simulations in TRNSYS, a 7 cm hull will be used, analyzing an enclosure that isolates more from the exterior, thus allowing the assessment of the influence of meteorological variables with larger enclosures than the actual ones.

In order to accurately simulate the thermal behavior of this structure, it is imperative to ascertain the ship’s displacement, which is defined as the product of its submerged volume (live work) and the specific weight of the surrounding fluid, representing the volume of water displaced. However, determining displacement solely based on load and fluid characteristics is not precise enough due to the ship’s history of modifications, such as repairs, machinery replacements, and additional plating in certain areas of the hull, leading to variations in its weight. To address this, drafts were measured in one of the rooms at the post, yielding a bow draught of 2.6 m and a stern draught of 2.7 m. The displaced volume of the vessel was determined using hydrostatic curves, which relate the vessel’s displacement and draft under various loading conditions. Hydrostatic curves were provided by the patrol manufacturer. Subsequently, a displacement value of 350 tons was derived from the average draft, which was crucial for conducting the thermal simulation. It must be considered that hydrostatic curves were employed without accounting for potential modifications specific to the Tabarca vessel, which could introduce variations in the precision of the simulations. Also, for simplification, there were no considered differences among live and dead volumes.

### 2.2. Thermal Model Definition

The patrol geometry was done with the Trimble SketchUp modeling software version 2022.0 (Figure 2). Regarding the simulation, a single thermal zone was defined. There were no considerable differences in the results between dividing the ship into several zones or considering it as a single zone. The patrol thermal model was created using the TRNSYS software (v17) [14] based on known patrol envelope characteristics (Table 1). EnergyPlus (v8.0) was used to create the weather-.epw format compatible with TRNSYS files from the data provided by the MeteoGalicia agency. The temporal resolution of data collected from both the established station and MeteoGalicia was set at one-hour intervals, aligning with the simulation’s time step. Standard meteorological datasets spanning a one-year period were produced, comprising 8760 individual records.

The model developed within the TRNSYS simulation framework (Figure 3) was constructed based on empirical data regarding vessel occupancy. All components utilized in the simulation were sourced from TRNSYS libraries. Each component within TRNSYS denotes a specific facet of the system under examination. To replicate the thermal dynamics of the vessel, TRNSYS Type 56 was utilized, defining the vessel as a singular temperature zone for procedural simplicity. Sensible heating and cooling demands were computed under ideal conditions, assuming perpetual availability of thermal reserves within the vessel. Given the vessel’s year-round operation, climate control systems must cater to space heating, ventilation, and summer cooling requirements. Monitoring activities spanned six months, during which setpoint temperatures were established (Figure 4). This involved the use of 12 iButton DS1923 sensors, offering a temperature accuracy of ±0.5 °C within the range of −10 °C to +65 °C. The ship does not have forced ventilation systems for air exchange.

### 2.3. Meteorological Variables

According to ASHRAE [15], six meteorological variables (air temperature, relative humidity, pressure, global horizontal irradiance, wind speed, and wind direction) are required for developing a thermal simulation model in buildings. For ship thermal simulations, the addition of seawater temperature to this set of variables is necessary. Seawater temperature significantly impacts the operational performance of ships, particularly affecting the submerged parts.

To assess the congruence between data collected by our developed in situ station and those obtained through the nearest neighbor method, we utilized information from the network of stations managed by the Galician Meteorology Agency (MeteoGalicia). This agency operates a network comprising 150 weather stations scattered across Galicia, located in northwestern Spain. While all stations record temperature and relative humidity, only 105 stations provide data on global solar radiation. Furthermore, 98 stations are equipped with pressure sensors, and 67 stations measure meteorological wind parameters. It is pertinent to highlight that the station nearest to the study area may lack certain variables that are crucial for conducting simulations. Additionally, MeteoGalicia maintains an oceanographic network consisting of marine stations. We obtained seawater temperature data from the buoy nearest to the port where the vessel under examination is anchored, specifically the buoy situated at the Rande Bridge in Vigo, positioned 12.22 km away from the vessel’s dock (Longitude: −8°39.6′ W, Latitude: 42°17.19′ N). The temperature sensor on the MeteoGalicia oceanographic network buoy has a measurement range of −40 °C to 105 °C and an accuracy of 1/3 DIN. It should be noted that in recent years, the MeteoGalicia agency has been using the values provided by the SAF OSI (Ocean and Sea Ice), which processes data from the MSG and GOES-East satellites, to indicate the temperature of the seawater.

The other weather variables were sourced from the station located in Lourizán, Pontevedra, which is approximately 4.07 km from the vessel’s location. This station (refer to Table 2) furnishes technical specifications akin to those of the weather station near the vessel’s home port.

### 2.4. In Situ Variables

The sensors employed in this study operate with a consistent sampling interval of one hour and data collection occurred on a weekly basis. Moreover, the occupancy rates of both active and inactive work zones within the vessel were ascertained. Temperature setpoint values delineated distinct heating (15.81 °C) and cooling (27.00 °C) periods. Notably, these criteria deviate from those typically applicable to residential settings [16]. It was supposed that the winter regime ran from the last Sunday of October to the first day of April. The thermal simulation took into account the sensitive load caused by internal lighting contributions. Internal gains from occupancy were computed using vessel use information and ISO 7730 [17]. ASHRAE’s empirical approach was used to calculate the vessel infiltration [15]. Upon data analysis, no discernible disparities were observed between weekdays and weekends concerning air conditioning usage.

The Arduino IoT weather station by SparkFun was used (see Figure 5) in our study, connected to the Arduino Cloud. Specifically, we utilized the Wimp assembly, a product developed by SparkFun [18], which serves as a cost-effective personal weather station, ensuring accessibility irrespective of financial constraints. This station integrates a weather shield with an electric Imp, facilitating the transmission of real-time weather data to the Wunderground community [19]. Presently, the Wunderground platform hosts over 250,000 stations worldwide. Table 3 presents the materials utilized in constructing the weather station. The implemented setup featured the Arduino UNO R3 serving as the base station, to which all sensors were connected. This board operates on the ATmega328P microcontroller and is powered by a battery via an AC to DC adapter, diverging from earlier models by excluding the FTDI series USB controller chip. Communication between components was facilitated using the I2C protocol. To gather weather data, the Sparkfun Redboard (Sparkfun Electronics, Niwot, CO, USA) was employed, interfacing with the Weather Shield and subsequently linking to the Imp via series strings. The Weather Shield, designed for Arduino, facilitates the measurement of luminosity and temperature. Additionally, it supports connection to sensors for wind speed and direction, rainfall, GPS, brightness, and humidity. Notably, the shield integrates the HTU21D humidity and temperature sensor (±2% accuracy), the MPL3115A2 barometric pressure sensor (±50 hPa accuracy), and the ALS-PT19 light sensor. These sensor accuracies are comparable to those found in government weather station equipment. The Weather Shield, compatible with HTU21D and MPL3115A2 Arduino libraries, provides two available RJ11 connector slots for attaching rainfall and wind sensors, along with a 6-pin connector for GPS.

The Electric Imp served as a Wi-Fi adapter, facilitating connectivity between devices and the internet via the Wi-Fi network (Figure 5). Integrated within the Electric Imp are components such as an 802.11b/g/n Wi-Fi transceiver, a sizable antenna, and a Cortex-M3 core. Programming for the Electric Imp is conducted using the Squirrel language. In this study, the Electric Imp was employed to interface with the RedBoard, receiving serial data which it then transmitted to a cloud-based agent. This agent, functioning as a cloud service, subsequently disseminated weather measurements online or stored them locally. Modification of the Electric Imp shield was required in three key aspects. A tower structure was made. The Arduino was placed at the bottom, the SparkFun Electric Imp Shield in the middle, and the SparkFun Weather Shield at the top (see the first figure in Section 3 for more information). The SparkFun Sunny Buddy was used to power the weather station, connecting the Solar Cell Large and the Polymer Lithium Ion Battery. The developed weather station is powered by a 3.7 V, 6000 mAh LiPo battery.

To safeguard temperature and relative humidity sensors, a custom-designed enclosure was devised (see Figure 6). Given the intended low-power nature of the system, a combination of battery and solar power sufficed. Seawater temperature was also assessed in situ, utilizing sensors identical to those employed for measuring onboard temperature. DS9107 waterproof capsules facilitated the submersion of these sensors. Global horizontal radiation was gauged with a pyranometer, specifically the CMP-3 model by Kipp & Zonen. This model corresponds to those utilized in the MeteoGalicia agency’s network and adheres to ISO 9060’s spectrally flat Class C standards [20]. The developed weather station was located in the harbor basin (pontoon) where the vessel under study is berthed (Figure 6). This type of surveillance vessel spends approximately 80% of its time moored in port, so locating the station on land in the vicinity of the ship seems advisable, as it simplifies the analysis. That is, the simulation was done assuming the ship was moored in port. Other military or surveillance vessels have the same usage patterns.

### 2.5. Performance Analysis

The validity of the assumptions used and the accuracy of the experimental parameter measurements are the two key factors that determine how reliable an experiment’s results are [21]. The average model performance error was analyzed in this work, as in most studies conducted on this subject, using various dimensional statistics [22,23,24]. The objective of experimental error measurements is generally to measure the precision and accuracy of multiple models. While random errors can cause the models to be imprecise, systematic errors during experimentation frequently lead to a lack of accuracy. As such, while comparing different interpolation techniques, both accuracy and precision error measurements should be considered. The results were examined in terms of mean absolute error (MAE) and mean bias error (MBE) to evaluate the accuracy of the estimations. Measurements of accuracy are the MBE and MAE. The MBE determines if a bias, or a systematic under- or overestimation of the model’s results, may occur. Positive MBE values show an overestimation. In contrast, an underestimation is indicated by negative values. Willmott and Matsuura’s advice was followed when applying the MAE to describe the average difference. A comparison of means analysis for independent samples (Student’s *t*-test) was also performed to determine the effects of the data source (onsite station vs. MeteoGalicia agency station) on the meteorological variables recorded (i.e., temperature, atmospheric pressure, relative humidity, global radiation, wind direction, wind speed, and seawater temperature).

## 3. Results and Discussion

### 3.1. Meteorological Data Comparison

Records from the two data sources were compared (i.e., onsite low-cost weather station and MeteoGalicia agency weather station). The actual meteorological values are temperature, relative humidity, pressure, global radiation, wind direction, and wind speed. The meteorological data from 2021 (time hourly counts) was analzed.

The performance of the data sources (weather stations) was statistically studied based on the data obtained. The results of the effects of the station type factor (self-developed on-site station vs. MeteoGalicia government agency nearest station) on the meteorological variables showed that temperature records on board the vessel (M = 14.94) are higher, t(17,503) = 16.05, *p* < 0.05, d = 0.53 (moderate effect size), than temperature records from the nearest station (M = 12.78); that relative humidity records on board the vessel (M = 82.25) are higher, t(17,491) = 29.91, *p* < 0.05, d = 0.64 (moderate effect size), than the relative humidity records of the nearest station (M = 74.33); that the shipboard atmospheric pressure records (M = 98,921.13) are higher, t(17,343) = −210.37, *p* < 0.05, d = 0.64 (high effect size), than the atmospheric pressure records of the nearest station (M = 96,403. 63); that the global horizontal radiation records on board the vessel (M = 173.03) are higher, t(17,354) = 6.38, *p* < 0.05, d = 0.26 (low effect size), than the global horizontal radiation records of the nearest station (M = 148.65); that the wind direction records on board the vessel (M = 181.91) are higher, t(16,892) = 2.64, *p* < 0.05, d = 0.23 (low effect size), than the wind direction records of the nearest station (M = 178.86); and that the shipboard wind speed records (M = 3.23) are higher, t(17,159) = 23.87, *p* < 0.05, d = 0.27 (low effect size), than the wind speed records of the nearest station (M = 2.88). Global horizontal radiation, wind direction, and wind speed records on the vessel also exhibited significant differences, indicating the impact of station selection on data accuracy.

To further validate the findings, seawater temperature data obtained from in situ monitoring over 6 months were compared with the data provided by the nearest buoy of the MeteoGalicia agency. The average temperature difference was observed to be 1.06 °C (SD 0.86), with variations in maximum and minimum temperatures of 0.28 °C (SD 0.49) and 1.99 °C (SD 1.43), respectively. Considering the high transmittance coefficient of the ship’s envelope, it becomes evident that relying on data from the nearest station significantly influences simulation results. Notably, seawater temperature monitoring was only feasible when the ship was stationary, underscoring the importance of context-specific data sources for accurate environmental simulations.

Table 4 shows the mean and CV values (from hourly records) for the six measured weather variables. From the results in Table 4 it can be deduced that there is a tendency to overestimate the variables temperature, relative humidity, and atmospheric pressure, while wind speed and direction and global horizontal radiation are underestimated. The variables with the greatest discrepancy are, in order: global radiation, temperature and wind speed. On the other hand, in atmospheric pressure and wind direction there is hardly any difference. Other authors also found wind measurement being a critical challenging aspect [25].

It should be noted that these are variables with a very low CV. In the case of temperature and global horizontal radiation, it is especially important since the average difference exceeds 10% and its influence on thermal behavior is greater than that of other variables (e.g., wind speed and direction), which will have consequences for the possible decisions taken based on this data. These analyses are supported by the results of the statistical variables showed in Table 4 (MBE and MAE). It is especially relevant that, using the same pyranometer as in the MeteoGalicia network, between two locations 4.07 km apart, there is, on average, more than 10% difference in this variable, with peaks exceeding 20%. Depending on the application of the meteorological data, these differences can be of great relevance. As expected, the pressure records at the vessel are higher than those collected at the nearest station since it is at a higher altitude. This tendency also occurs in the relative humidity records, where the MeteoGalicia agency network station shows lower values than those recorded by the station located at sea level. In relation to pressure measurements, the values recorded by the developed station, on average, were higher than those recorded by the MeteoGalicia governmental agency network station. This all indicates the relevance of measuring in situ in unique locations, such as at sea level, where governmental weather station networks are rarely deployed. The range of values of the records of the six meteorological variables shown in Table 4 also shows the discrepancies between the records of the two weather stations. For example, the minimum value of relative humidity recorded by the developed station was 25%, while the MeteoGalicia network station had a minimum of 6% in the same period. The same is true for the rest of the variables. The discrepancies in the wind direction data are especially evident. However, for the application in which we will implement the variables in this study, this will not have a significant impact.

The statistical distribution of the meteorological variables for the two sources is shown using a violin plot (Figure 7). A box plot and a symmetric kernel density plot are combined to create the violin plot. This allows for the comparison of distributions by providing precise data on the distribution’s spread, outliers, asymmetry, and central values [26,27].

The density of data determined using the kernel approach is represented by the shape (Figure 8). More data are connected to a given value when the shape is wider. The thin line represents the lower and upper percentiles, while the thick line segment displays the 25th to 75th percentile. Graphs showed the abovementioned overestimation for temperature, atmospheric pressure, and relative humidity, which highlights the notable challenge of accurately measuring environmental variables. Particularly striking is the significant dispersion in wind direction recorded at the in situ station; the nearest station, situated at the port near the sea without any structural interference, revealed a more accurate representation of wind patterns. However, complexity arises from the orographic disparities between the MeteoGalicia agency’s network station and the vessel’s location. Attempting to reconcile data from a nearby station becomes problematic due to the distinct geographical and orographic characteristics, introducing errors in the analysis. This underscores the importance of considering the specific context and local conditions when relying on weather data, emphasizing the need for precise measurements tailored to the unique environmental features of the study area.

The measurement equipment’s installation and positioning significantly influence the data’s accuracy. On ships, spatial constraints and operational requirements often dictate the placement of sensors, which may only sometimes be ideal from a meteorological data collection standpoint. For instance, sensors might be placed in locations that are subject to heat emissions from the ship or are shielded from prevailing wind patterns, skewing temperature and wind speed readings. In contrast, meteorological stations on shore are usually optimally positioned to minimize such impacts, adhering to standard protocols that aim to ensure data reliability. The weather station developed was located near the ship, at sea level in the dock, but not on the ship. This was done so as not to have to consider the ship’s movements.

Furthermore, standard meteorological measuring equipment is housed inside the Stevenson screen to keep out precipitation and direct heat radiation from outside sources while enabling air to flow around it. The temperature and humidity sensors distributed throughout the ship, which are also used to monitor seawater temperature, were placed in IP68 bags to protect them from dust and water. This justifies the consistent differences found between the measurements of the different devices. Additionally, the variations between the data obtained from in situ measurements and those recorded by shore-based meteorological stations can partly be attributed to the differential sensitivity and accuracy of the measurement instruments used in each setting. Instruments deployed in maritime environments must endure harsher conditions than their land-based counterparts, which may affect their sensitivity and calibration over time.

### 3.2. Influence on Thermal Simulations

In this research, the results of the thermal demands were compared to demonstrate the relevance of measuring in situ versus taking data from the nearest weather station. The Type56 (multizone dwelling), which was used to simulate the thermal behavior of the vessel, outputs the thermal zone loads (Qsens, https://www.qsens.io/). That is, Qsens is the sensible energy demand. Negative values of Qsens indicate heating loads, while positive values indicate cooling loads. The yearly energy usage, as well as the needs for heating and cooling, were computed. Table 5 presents an insightful comparison of how meteorological data from two different datasets impacts the monthly heating and cooling loads (kWh) for the vessel.

Notably, the variations in peak values are more pronounced when analyzed at daily or hourly intervals as opposed to monthly aggregations. The results reveal a distinct seasonal pattern, with heating needs prevalent for most of the year, while cooling demands are observed from 15th June to 31st August. The simulation based on onsite weather data indicates an annual total consumption of 4668.29 kWh (46.68 kWh/m^2^) for heating and cooling (Figure 8). Table 5 further underscores the potential for up to a 9% difference in monthly and yearly simulated energy usage outcomes depending on the data source. These significant disparities emphasize the substantial impact of the chosen weather data source on heating and cooling demands in building simulations.

The study discerns those various meteorological characteristics, including air temperature, pressure, relative humidity, global horizontal irradiance, wind direction, wind speed, and seawater temperature, exert differential effects on the building simulation. The thermal simulation specifically relies on air temperature for calculating exterior vessel temperature and infiltrations, with deviations at high-temperature bin levels having a more pronounced impact on cooling energy needs than global horizontal irradiance at low-temperature bin levels [28]. Infiltrations are computed based on wind speed, pressure, and relative humidity, following ASHRAE guidelines [15,29]. The seawater temperature notably influences the submerged part of the vessel, emphasizing the contextual importance of each parameter in the simulation process. It is crucial to highlight that the magnitude of deviations in the utilized data’s influence depends on the specific application. In this case study, the distinct internal setpoint temperatures for activating the air-conditioning system played a pivotal role, mitigating potentially significant differences in the estimated energy requirements resulting from diverse meteorological data sources.

## 4. Conclusions

This research underscores the critical importance of locally collected weather data for accurate simulation of shipboard thermal behavior. A patrol vessel on the Galician coast (NW Spain), named Tabarca, was chosen as the subject study. The patrol thermal model was created using the TRNSYS software, and meteorological datasets (air temperature, relative humidity, pressure, global horizontal irradiance, wind speed, and wind direction) for one year period were recorded. The congruence of in situ measurements with those obtained from the Galician Meteorology Agency (MeteoGalicia) was analyzed.

Discrepancies between on-site weather station data and records from nearby government agencies emphasize the necessity for in situ measurements for calibration. Analyses showed that there is a tendency to overestimate temperature, relative humidity, and atmospheric pressure, while wind speed and direction and global horizontal radiation resulted in an underestimation. This overestimation may reach up to 20% and an average deviation of 10%. The comparison with the nearest station, situated at the port near the sea without any structural interference, revealed a more accurate representation, particularly for wind patterns. The orographic disparities between the local agency network station and the in situ measurements revealed important errors in the data obtained.

The application of meteorological data to thermal simulations provides valuable insights into the energy dynamics of ships, revealing nuanced interactions between external environmental factors and internal thermal processes. The observed discrepancies in heating and cooling demands highlight the potential consequences of data imprecision, emphasizing the crucial role of accurate data in informing design decisions and operational strategies for maritime vessels. It is crucial to recognize the unique climatic conditions in maritime environments and the need for data from measurement buoys to accurately represent them. The research demonstrates deviations of over 30% when using nearby stations, emphasizing the inadequacy of relying on such data sources. The development and implementation of low-cost weather stations offer promising opportunities for enhancing data collection in maritime contexts, especially in areas where governmental weather stations may be limited.

The study advocates for the utilization of accessible and affordable meteorological monitoring for comprehensive data acquisition, highlighting the superiority of accurate in situ data even when recorded with low-cost sensors. Notably, in situ seawater temperature measurements revealed significant differences compared to data from nearby buoys, emphasizing the importance of directly measuring environmental conditions specific to the ship’s location. These findings have significant implications for improving energy efficiency in ship design and modernization. Locally collected meteorological data for calibration is crucial for reliable simulation results, and the study emphasizes the importance of considering unique factors such as vessel occupancy patterns and environmental conditions in maritime environments. Careful attention to the accuracy and relevance of meteorological data in thermal simulations is necessary for the effective planning and efficient operation of maritime structures.

Moreover, this study acknowledges certain limitations that could influence the precision of our thermal simulations. Specifically, using displacement data from the patrol manufacturer’s hydrostatic curves without adjustments for potential modifications specific to the Tabarca vessel could affect accuracy. This oversight suggests that alterations in the vessel’s structure due to repairs or modifications could lead to variances in simulation outcomes. Additionally, simplifying thermal gains attributed to occupancy into two broad categories must fully encapsulate the complex thermal dynamics experienced in real-world scenarios. This binary classification overlooks the nuanced temperature variations throughout the vessel.

In simulating the vessel’s thermal behavior, it was treated as a static entity within software primarily designed for buildings, disregarding its dynamic nature during navigation. Such an approach could result in inaccuracies, particularly concerning changing solar exposure and wind conditions. Furthermore, assumptions of a fixed live volume within the vessel, without accounting for fluctuations due to operational activities, such as refueling or provisioning, could further distance simulation results from real-life conditions.

Compiling meteorological data from varied sources without specific consideration of local anomalies or temporal variations, alongside the failure to account for shadow effects from nearby structures, may have compromised the radiation simulation’s fidelity. Attempts to collect in situ meteorological data faced challenges due to the location’s adverse climatic conditions and the limitations of the equipment setup, leading to unreliable temperature and wind data for use in the vessel’s thermal simulation. By adopting generic insulation values and treating live and dead volumes as equivalent, the generalization of material properties may not truly reflect the vessel’s distinct thermal characteristics and behaviors, thus affecting the simulation’s accuracy. These limitations underscore the importance of detailed, vessel-specific data and the consideration of dynamic environmental interactions for enhancing the reliability of thermal simulations in maritime contexts.

## Figures and Tables

**Figure 1 sensors-24-02454-f001:**
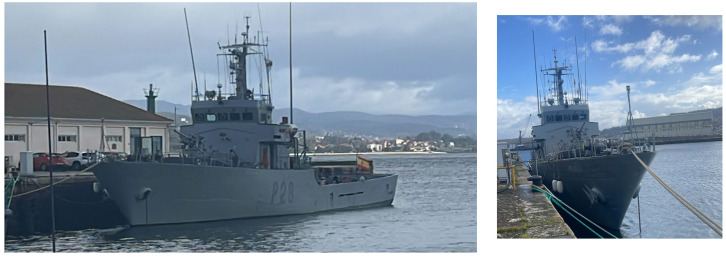
Tabarca patrol moored in port.

**Figure 2 sensors-24-02454-f002:**
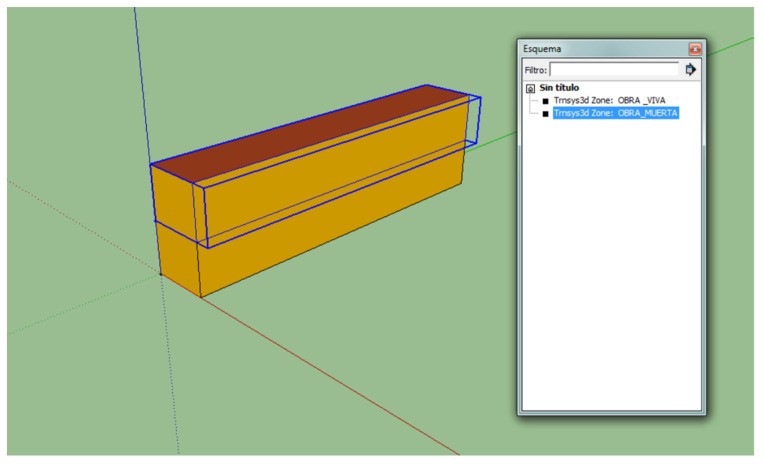
Patrol model in SketchUp. “Obra viva” is underwater hull; “Obra muerta” is upper works.

**Figure 3 sensors-24-02454-f003:**
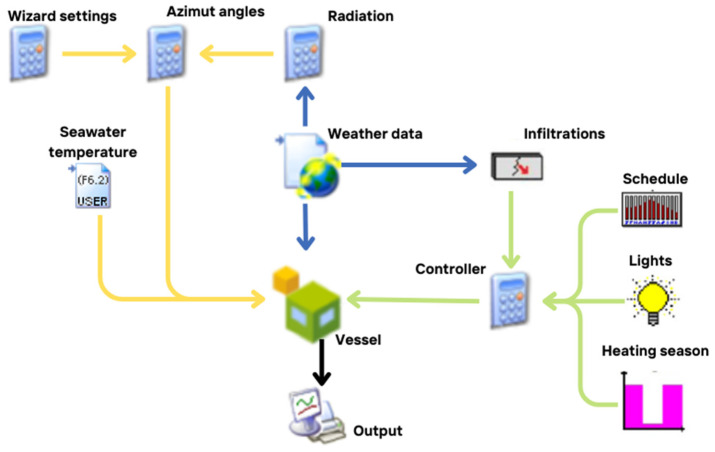
Simulation panel from TRNSYS.

**Figure 4 sensors-24-02454-f004:**
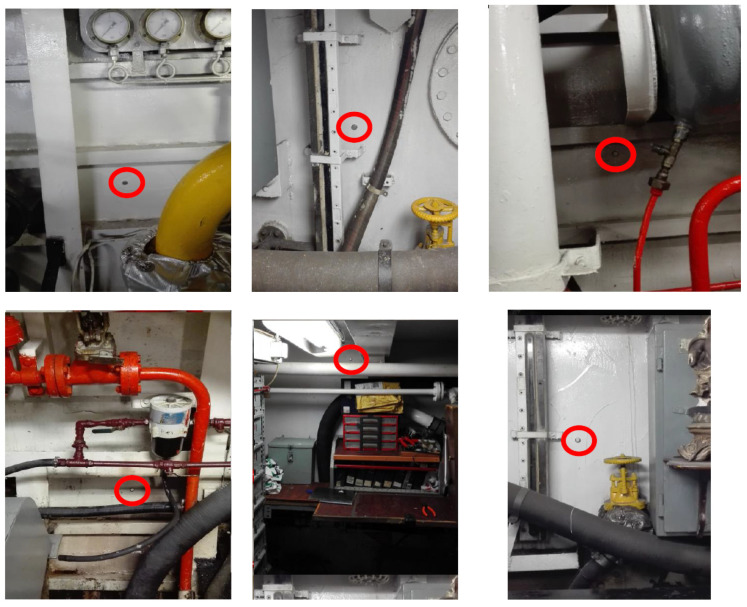
Temperature sensors in the engine room and storeroom.

**Figure 5 sensors-24-02454-f005:**
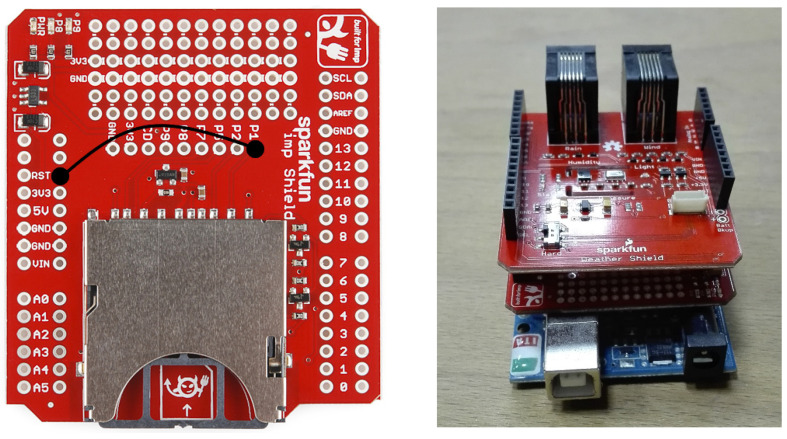
(**left**) Electric Imp shield (**right**) final assembly.

**Figure 6 sensors-24-02454-f006:**
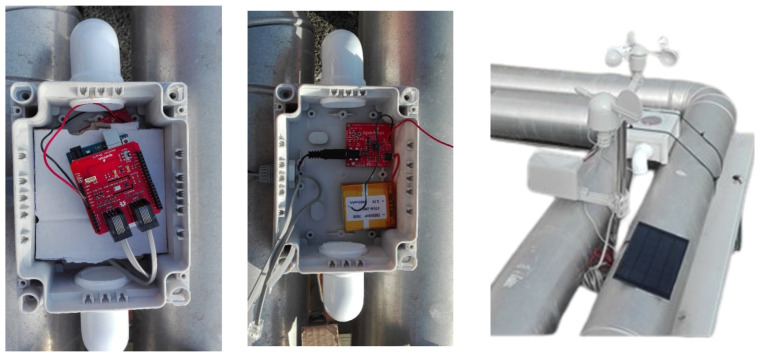
Distribution of the shields inside the shell and complete weather station.

**Figure 7 sensors-24-02454-f007:**
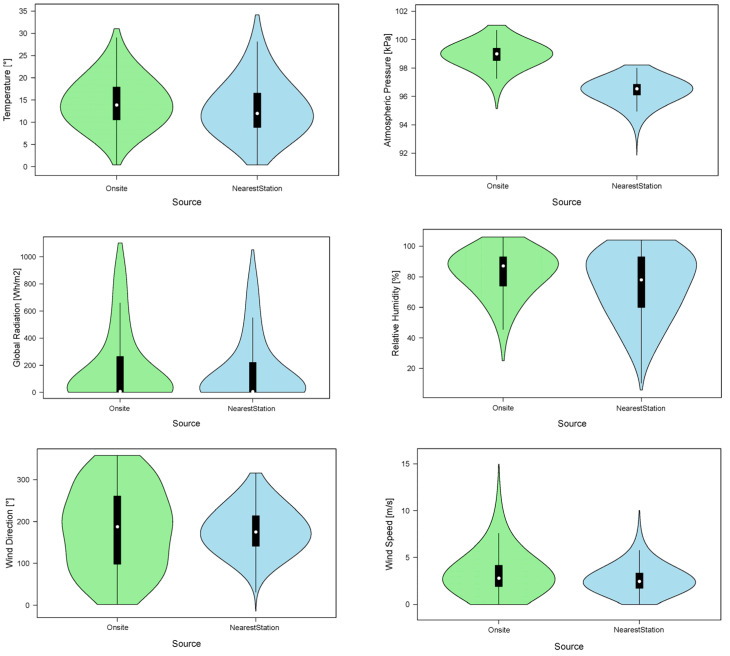
Violin plot for weather variables.

**Figure 8 sensors-24-02454-f008:**
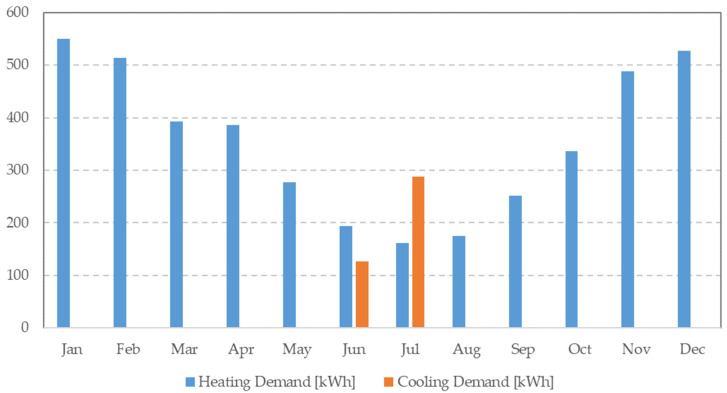
Monthly heating and cooling demands in kWh (hourly accounts) based on onsite weather data.

**Table 1 sensors-24-02454-t001:** Properties of each enclosure type.

Layer Type	Thermal Conductivity [W/m·K]	Capacity [kJ/kg·K]	Density [kg/m^3^]	Thickness [m]
Steel	18.1	0.5	7800	0.007
Insulating material	0.14	1.0	50	0.050

**Table 2 sensors-24-02454-t002:** Sensors used at the weather station placed near the patrol vessel’s home port (Marin Port) and its accuracy.

Measured Parameter	Sensor Used	Accuracy
Wind speed	Campbell 05106-5 MA (Campbell scientific, Logan, UT, USA)	±0.3 m/s
Wind direction	Campbell 05106-5 MA (Campbell scientific, Logan, UT, USA)	±3°
Barometric pressure	Vaisala PTB110 (Vaisala, Vantaa, Finland)	±0.3 hPa at +20 °C
Global horizontal radiation	Kipp&Zonen CMP-3 (Kipp&Zonen, Delft, The Netherlands)	20 µV/W/m^2^
Temperature	Rotronic HygroClip HC2A-S3 (Rotronic AG, Bassersdorf, Switzerland)	±0.1 °C
Relative humidity	Rotronic HygroClip HC2A-S3 (Rotronic AG, Bassersdorf, Switzerland)	±0.8%

**Table 3 sensors-24-02454-t003:** Weather station required materials.

Electronics	Function
SparkFun RedBoard– Programmed with Arduino	Gather all the weather data
Electric Imp Shield	Connect hardware device
SparkFun Weather Shield	Measure barometric pressure, relative humidity, luminosity, and temperature
Shield Headers	Connect shield to Arduino board
RJ11 Connectors	Provide an input signal
Weather meter	Measure wind speed, wind direction and rainfall
Solar panel	Provide power
Lithium Ion Battery—6Ah	Storage energy
SparkFun Sunny Buddy— MPPT Solar Charger	Connect solar panel to battery

**Table 4 sensors-24-02454-t004:** Mean annual values for the weather variables and statistical summary.

Variables	On-Site Station				MeteoGalicia Agency Nearest Station				MBE ^c^	MAE ^d^
	*M* ^a^	*CV* ^b^	Range		*M*	*CV*	Range			
			Min.	Max.			Min.	Max.		
Temperature [°C]	14.94	0.33	3	30	12.78	0.51	2	29	−1.32	3.79
Relative humidity [%]	82.25	0.17	25	106	74.33	0.28	6	104	−7.92	19.42
Atmospheric pressure [hPa]	989	0.01	951	101	964	0.01	918	982	−2.57	2.54
Global horizontal radiation [Wh/m^2^]	173.03	1.55	0	1102	148.65	1.58	0	1052	24.38	83.63
Wind direction [°]	181.91	0.53	2	358	178.86	0.28	−14	316	3.04	85.97
Wind speed [m/s]	3.23	0.65	0	11	2.88	0.54	0	10	0.64	1.90

^a^ M: Mean value, ^b^ CV: $Coefficient\;of\;variation=\frac{Standard\;deviation}{Mean}$, ^c^ MBE: $Mean\;Bias\;Errror=\sum_{i=1}^{n}\frac{X_{i}-Y_{i}}{N}$, ^d^ MAE: $Mean\;Absolute\;Errror=\sum_{i=1}^{n}\frac{\left|X_{i}-Y_{i}\right|}{N}$.

**Table 5 sensors-24-02454-t005:** Heating and cooling vessel demands.

		On-Site Station	MeteoGalicia Agency Station
Monthly demands Heating + Cooling [kWh]	January	549.85	557.82
	February	513.73	521.12
	March	393.18	406.42
	April	386.01	401.45
	May	276.83	284.73
	June	320.54	340.48
	July	449.92	490.40
	August	174.71	189.25
	September	251.49	268.08
	October	335.66	348.63
	November	488.69	497.99
	December	527.68	534.21
Annual demands	Total annual energy demand [kWh]	4668.29	4840.60
	Annual energy demand [kWh/m^2^]	46.68	48.41
	Total annual difference [kWh]		172
	Annual difference [kWh/m^2^]		2
	Annual percentage difference		3.69%

## Data Availability

Data are contained within the article.

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
