# Peer review of "Comparative Analysis of Meteorological versus In Situ Variables in Ship Thermal Simulations"

_sensors, 2024, doi:10.3390/s24082454_

Round 1
Reviewer 1 Report
Comments and Suggestions for Authors
This article explores the influence of meteorological variables on thermal simulation on ships, compares on-site measurements with meteorological variable data, and identifies the relative advantages of each method in improving simulation fidelity. This article has a clear goal and clear logic, and through a combination of graphics and text, readers can quickly understand its research methods and processes. The author also provided a detailed explanation of the professional symbols in the research data for easy reading.
The advantages of this article are as follows:
1.At present, most thermal simulations are idealized environments, and this technology is rarely applied in the maritime field. The author pays attention to the influence of meteorological variables on thermal simulations and chooses the ship environment for research, which is the breakthrough and innovation point of this study.
2. This study was conducted in a real ship environment, but the author considered many hidden factors that may have an impact, which improved the accuracy of this study.
3. The article contains many charts, allowing readers to clearly see different data results for comparison.
Here are some small suggestions:
1. It is recommended to add some research results from recent years as references.
2.The conclusion section only has textual explanations, which can be further elaborated in conjunction with the research results.
3. After the third title, It jumps directly to the fifth title, and there are two identical sections 2.4 in the text. Please modify the numbering issue.
4. The description in section 3.1 is slightly more complex. Can it be explained through the order of meteorological variables or other logical sequences in the text.
5. In the 2.1 ship model, some irrelevant ship introduction sections can be reduced, and important data can be specifically explained on how to obtain an average draft to obtain a displacement of 350 tons.
6. This study is conducted around a real environment simulation, and it would be best to further explain some of the limitations of this study.
7. Can you add some opportunities or reasons for studying this patrol ship.
Comments on the Quality of English Language
Minor editing of English language required
Author Response
Review report 1: comments to revisions
This article explores the influence of meteorological variables on thermal simulation on ships, compares on-site measurements with meteorological variable data, and identifies the relative advantages of each method in improving simulation fidelity. This article has a clear goal and clear logic, and through a combination of graphics and text, readers can quickly understand its research methods and processes. The author also provided a detailed explanation of the professional symbols in the research data for easy reading.
Answer: Thank you very much for your comments. In this manuscript we have addressed the study of thermal simulations in the specific marine environment and how meteorological variables affect them. We believe that this manuscript will contribute to increase knowledge incorporating real-time meteorological data and vessel measurements enhancing simulation accuracy and highlighting the importance of calibration for precise energy efficiency assessments.
The authors would like to thank the comments made by the reviewer, which will help us to improve the overall quality of the manuscript and that we have considered in this review. In order to facilitate review, we would like to indicate you that changes in the manuscript after your suggestions have been highlighted in green (other colours correspond to the other reviewers).
Specific comments and answers:
- At present, most thermal simulations are idealized environments, and this technology is rarely applied in the maritime field. The author pays attention to the influence of meteorological variables on thermal simulations and chooses the ship environment for research, which is the breakthrough and innovation point of this study.
Answer: Thank you for your insightful comments regarding our study on thermal simulations in the maritime domain, particularly focusing on ships. We appreciate the opportunity to respond and provide further clarification on the significance of our research. It is indeed acknowledged that thermal modelling has become indispensable in predicting the energy requirements of buildings, and we echo the sentiment that the efficacy of these models under idealized conditions can sometimes be met with scepticism. Our study endeavors to address this skepticism by extending the application of thermal simulations to the maritime field, where such models are currently underrepresented.
Our study extends this paradigm to ships, where we explore the intricate interactions within and around the ship structure, incorporating factors like construction attributes, HVAC systems, and external elements including weather conditions. Furthermore, we recognize the importance of integrating localized meteorological data to enhance the fidelity of energy simulations. In our research, we specifically focus on the challenges posed by ship simulations, including the precision of meteorological data for offshore applications. Through our study, we aim to contribute to the advancement of methodologies for accurately evaluating the energy demand of ships via thermal simulations.
While acknowledging the existing literature gap in applying advanced thermal simulation techniques to the maritime domain, our study presents a novel approach by evaluating the performance of low-cost and ad-hoc sensors for energy calibration within the TRNSYS simulation framework. By comparing data obtained from these sensors with observations from automatic weather stations, we seek to provide insights into the efficacy of meteorological variables in thermal simulations for ships.
We appreciate your valuable feedback and assure you that your suggestions will be considered in refining our study.
- This study was conducted in a real ship environment, but the author considered many hidden factors that may have an impact, which improved the accuracy of this study.
Answer: Our study on thermal simulations in real ship environments prioritizes accuracy by meticulously addressing critical factors. To ensure comprehensive simulation accuracy, we collect detailed local meteorological data, including air temperature, humidity, solar radiation, wind speed, and seawater temperature. These efforts reflect our commitment to capturing the complexity of maritime conditions and delivering reliable results. By considering these hidden factors, we enhance the fidelity of our thermal simulations, facilitating better understanding and decision-making in ship design and operation.
- The article contains many charts, allowing readers to clearly see different data results for comparison.
Answer: Thank you very much for your comments. We have tried to make the manuscript the most comprehensible possible to readers and contributing to advances in the field of thermal simulation in maritime applications, bridging a gap in research and paving the way for more effective energy management and operational optimization strategies in the maritime sector.
- It is recommended to add some research results from recent years as references.
Answer: Following reviewer suggestions, 3 references within the last five years have been added to the manuscript:
Coraddu A., Gil A., Akhmetov B., Yang L., Romagnoli A., Ritari A., Huotari J., Tammi K. (2022). Energy storage on ships. Sustainable Energy Systems on Ships: Novel Technologies for Low Carbon Shipping, pp. 197 – 232. DOI: 10.1016/B978-0-12-824471-5.00012-8.
Karachalios T., Moschos P., Orphanoudakis T. (2024). Maritime Emission Monitoring: Development and Testing of a UAV-Based Real-Time Wind Sensing Mission Planner Module. Sensors, 24 (3), art. no. 950. DOI: 10.3390/s24030950.
Achite M., KatipoÄŸlu O.M., Javari M., Caloiero T. (2024). Hybrid interpolation approach for estimating the spatial variation of annual precipitation in the Macta basin, Algeria. Theoretical and Applied Climatology, 155 (2), pp. 1139 – 1166. DOI: 10.1007/s00704-023-04685-w.
- The conclusion section only has textual explanations, which can be further elaborated in conjunction with the research results.
Answer: Following reviewer suggestions, the conclusions section has been re-written to include more details of the materials and methods and also the results obtained (page 13-14, lines 400-414).
- After the third title, it jumps directly to the fifth title, and there are two identical sections 2.4 in the text. Please modify the numbering issue.
Answer: Numbering has been corrected throughout the text. So, the mistake in the second 2.4 section has been corrected and changed to “2.5. Performance analysis” (page 9, line 246). Conclusions section has been also re-numbered to “4. Conclusions” (page 13, line 398).
- The description in section 3.1 is slightly more complex. Can it be explained through the order of meteorological variables or other logical sequences in the text.
Answer: Following the reviewer suggestions, the 3.1 section has been re-written (pages 9-11).
- In the 2.1 ship model, some irrelevant ship introduction sections can be reduced, and important data can be specifically explained on how to obtain an average draft to obtain a displacement of 350 tons.
Answer: In section “2.1 Ship description” some details, related to weekly missions, crew capacity and others have been deleted and more information about the volume displacement has also been added (page 3, lines 110-113): “The displaced volume of the vessel was determined using hydrostatic curves, which relates the vessel's displacement and draft under various loading conditions. Hydrostatic curves were provided by the patrol manufacturer”.
- This study is conducted around a real environment simulation, and it would be best to further explain some of the limitations of this study.
Answer: Following reviewer suggestions some of the limitations have been added to the text. So, in page 3 (lines 125-127) an explanation of hydrostatic curves employed to determine displacement volume has been included. Hydrostatic curves were provided by the manufacturer, and they are the current form to determine volume although they do not take into account potential modifications of the vessel since manufacturing and this could introduce some variations in the measured parameters. Also, we cannot discard that there may be differences between live and dead volume and they have not been considered in this study although we would like to remark that no navigation was recorded during the measurements period.
- Can you add some opportunities or reasons for studying this patrol ship.
Answer: Additional information justifying the selection of the vessel under study has been incorporated to the text (page 3, lines 102-112):
“The Tabarca patrol belongs to the Spanish Navy, and it is used for navigation and occasionally made available for research tasks. Its construction details are known, including its enclosures, which regulate the flow of external air, the incidence of solar radiation, the entry of natural light, and the transmission of heat (both inward and outward). The vessel's materials and geometry are also documented, and it is constructed with fully welded A-grade naval steel measuring 44.5 meters in length and 6.6 meters in beam. The Tabarca's hull consists of 7-millimeter-thick A-grade steel coated internally with a 5-centimeter layer of thermal insulation known as rock wool. For simulations in Trnsys, a 7-centimeter hull will be used, analyzing an enclosure that isolates more from the exterior, thus allowing the assessment of the influence of meteorological variables with larger enclosures than the actual ones”.
Reviewer 2 Report
Comments and Suggestions for Authors
The paper is interesting and original. Unluckily, the authors only describe the differences between meteorological and in-situ variables and estimate their impact on ship thermal simulations. The presented analyses could be more interesting if the authors also discuss the potential causes of these differences.
Differences may result from natural causes resulting from the fact that one is comparing measurements carried out in two different locations, but also from 3 additional reasons:
1. There are differences in the sensitivity and accuracy of the measurement equipment used
2. The way the equipment is installed affects its readings,
3. The ship-mounted station moves, but the shore stations do not
Ad 1) The article provides the technical specifications of the ship's meteorological station; the parameters of measuring devices used at shore stations and buoys are unknown.
Ad2) Instruments for typical meteorological measurements are located within the Stevenson screen to protect them against precipitation and direct heat radiation from outside sources while allowing air to circulate freely around them. The instruments onboard the patrol are placed in a small plastic enclosure exposed to sun (the air exchange system is not described – forced? natural?). This can explain why the air temperature measured on site is approximately 2K higher than in the meteorological network. The higher pressure measured on-site can result from how the enclosure is ventilated. The installed sensor can measure total pressure (the sum of static and dynamic pressure). Moreover, the height of the wind instrument placement impacts readings. Simply, wind speed changes with height above land/sea level.
Ad3) The fact that the patrol ship (on-site station) can move causes the wind station to measure apparent wind speed and direction, not a true value. Apparent wind is a product of adding vectors of true wind and boat speed. Many navigation systems, even for sport sailing yachts, allow navigators to select between true wind and apparent wind display. This requires the connection of GPS data or log data with the wind instruments. The cruising speed of the patrol 7-8 knots (?) can considerably affect the data related to wind measurements. However, it should be noted that apparent wind speed and direction are more appropriate for the energy calculations. Log data could be used to more accurately estimate the heat transfer coefficient of the underwater part of the patrol (convection to water is affected by boat speed).
Editorial comments
Table 4. The units of atmospheric pressure are wrongly presented as [%] instead of [Pa].
Generally, many data in the paper are presented with too many significant figures. For instance, the mean atmospheric pressure in Table 4 is presented with 7 significant figures (even using hundredths of Pascal) while the accuracy of Vaisala PTB110 is 30 Pa at +20 °C.
Author Response
Review report 2: comments to revisions
The paper is interesting and original. Unluckily, the authors only describe the differences between meteorological and in-situ variables and estimate their impact on ship thermal simulations. The presented analyses could be more interesting if the authors also discuss the potential causes of these differences.
Answer: Thank you very much for your comments. We have taken into consideration the comments made by the reviewers and try to address them. Detailed comments are provided below. First, we would like to thank the comments made by the reviewer, which will help us to improve the overall quality of the manuscript and that we have considered in this review. To facilitate review, we would like to indicate you that changes in the manuscript after your suggestions have been highlighted in yellow (other colours correspond to the other reviewers).
Specific comments:
- Differences may result from natural causes resulting from the fact that one is comparing measurements carried out in two different locations, but also from 3 additional reasons:
- There are differences in the sensitivity and accuracy of the measurement equipment used
- The way the equipment is installed affects its readings,
- The ship-mounted station moves, but the shore stations do not
Answer:
Based on the reviewer's comment, the possible reasons for the differences between the developed weather station and the MeteoGalicia network station were justified. This information has been added in section 3.1.
- Ad 1) The article provides the technical specifications of the ship's meteorological station; the parameters of measuring devices used at shore stations and buoys are unknown.
Answer:
Information on the temperature sensor of the MeteoGalicia network buoy used was included in section 2.3.
- Ad2) Instruments for typical meteorological measurements are located within the Stevenson screen to protect them against precipitation and direct heat radiation from outside sources while allowing air to circulate freely around them. The instruments onboard the patrol are placed in a small plastic enclosure exposed to sun (the air exchange system is not described – forced? natural?). This can explain why the air temperature measured on site is approximately 2K higher than in the meteorological network. The higher pressure measured on-site can result from how the enclosure is ventilated. The installed sensor can measure total pressure (the sum of static and dynamic pressure). Moreover, the height of the wind instrument placement impacts readings. Simply, wind speed changes with height above land/sea level.
Answer:
In response to this reviewer's comment, new information was added in sections 2.2 and 3.1.
- Ad3) The fact that the patrol ship (on-site station) can move causes the wind station to measure apparent wind speed and direction, not a true value. Apparent wind is a product of adding vectors of true wind and boat speed. Many navigation systems, even for sport sailing yachts, allow navigators to select between true wind and apparent wind display. This requires the connection of GPS data or log data with the wind instruments. The cruising speed of the patrol 7-8 knots (?) can considerably affect the data related to wind measurements. However, it should be noted that apparent wind speed and direction are more appropriate for the energy calculations. Log data could be used to more accurately estimate the heat transfer coefficient of the underwater part of the patrol (convection to water is affected by boat speed).
Answer:
The weather station was located in the immediate vicinity of the ship. This aspect was emphasized in sections 2.4 and 3.1. This simplified the analysis and processing of the data. It should be noted that in this type of ship, as is the case with many military naval units, 80% of the time they are stationary.
- Table 4. The units of atmospheric pressure are wrongly presented as [%] instead of [Pa].
Answer: Atmospheric pressure units have been corrected in table 4 (page 10).
- Generally, many data in the paper are presented with too many significant figures. For instance, the mean atmospheric pressure in Table 4 is presented with 7 significant figures (even using hundredths of Pascal) while the accuracy of Vaisala PTB110 is 30 Pa at +20 °C.
Answer: Values of atmospheric pressure in Table 4 had been expressed in Pa and they have now been expressed in hPa.